

# Factors of influence on flood risk perceptions related to Hurricane Dorian: an assessment of heuristics, time dynamics and accuracy of risk perceptions

Laurine A. de Wolf 1, Peter J. Robinson 1, Wouter J.W. Botzen 1, Toon Haer 1, Jantsje Mol 2 , Jeffrey Czajkowski 3

[1] Institute for Environmental Studies – Vrije Universiteit Amsterdam, 1081 HV Amsterdam, The Netherlands
[2] Center for Research in Experimental Economics and Political Decision Making (CREED), University of Amsterdam, Amsterdam, The Netherlands
[3] Center for Insurance Policy and Research, National Association of Insurance Commissioners (NAIC), Kansas City, USA

*Correspondence to:* Laurine de Wolf (l.a.de.wolf@vu.nl)



**Abstract.** Flood damage caused by hurricanes is expected to rise globally due to climate and socio-economic change. Enhanced flood preparedness among the coastal population is required to reverse this trend. The decisions and actions taken by individuals are thought to be influenced by risk perceptions. This study investigates the determinants that shape flood risk perceptions, as well as the factors that drive flood risk misperceptions of coastal residents. We conducted a survey among 871 residents in flood-prone areas in Florida during a five-day period in which the respondents were threatened to be flooded by Hurricane Dorian. This approach allows for assessing temporal dynamics in flood risk perceptions during an evolving hurricane threat. Among 255 of the same households, a follow-up survey was conducted to examine how flood risk perceptions vary after Hurricane Dorian failed to make landfall in Florida. Our results show that the flood experience and social norms have the most consistent relationship with flood risk perceptions. Furthermore, participants indicated that their level of worry regarding the dangers of flooding decreased after the near-miss of Hurricane Dorian, compared to their feelings of worry during the hurricane event. Based on our findings, we offer recommendations for improving flood risk communication policies.

**Keywords**

Flood risk perception; hurricanes; near-miss events





## 1. Introduction

Florida is one of the most at risk states in the United States for hurricanes (Basolo et al., 2017; Klotzbach et al., 2018). Hurricanes such as Katrina in 2005, Sandy in 2012, and Ian in 2022 resulted in catastrophic losses (Bostrom et al., 2018; Conroy, 2022). These losses from hurricanes are rising due to population and economic growth, and potentially climate change (Coronese et al., 2019; Knutson et al., 2019; Webster et al., 2005). Given the fact that climate change may increase the frequency of floods induced by hurricanes , residents' efforts to protect themselves and reduce their losses are crucial. Risk reduction strategies, such as evacuation and floodproofing measures are important responses to a hurricane threat to avoid damages and loss of life (Basolo et al., 2017; Botzen et al., 2019).

Given rising hurricane risk, one would expect an increase in hurricane preparedness activities. However, many households are currently underprepared for natural hazards (Basolo et al., 2009; Murti et al., 2014), which may be due to a low perception of risk (Dash & Gladwin, 2007; Lindell & Perry, 2012; Peacock et al., 2005). Moreover, individual perceptions of risk are often at odds with expert estimates of risk (Duží et al., 2017), with some individuals underestimating their risk and others overestimating the risk (Dueñas-Osorio et al., 2012). It is useful to understand how individual flood risk perceptions compare with expert risk assessments, as well as the factors influencing these perceptions, to improve flood risk communication strategies and flood risk management policies (Brown & Damery, 2002; Bradford et al., 2012; Senkbeil et al., 2019). For instance, policy makers can adapt current risk communication strategies to enhance support for flood risk reduction measures among the public (Bradford et al., 2012; Peacock et al., 2005).

Most prior analyses of flood risk perceptions associated with a hurricane threat rely on data collected at a single moment using cross-sectional surveys conducted after a hurricane has occurred (Basolo et al., 2017; Burnside et al., 2007; Demuth et al., 2016; Lechowska, 2018; Matyas et al., 2011). However, such an approach may not give adequate insights into risk perceptions during a hurricane threat. Risk perceptions may also vary after the hurricane event, depending on the severity of the experienced impacts. Understanding these dynamics regarding risk perceptions is important since many emergency hurricane preparations are made shortly before a hurricane makes landfall. Additionally, it is often observed that structural adjustments to properties to limit future disaster damage are made shortly after a disaster (Bubeck et al., 2012a). Both emergency preparedness actions taken during a threat and structural damage mitigation actions taken afterwards are likely to be guided by individual risk perceptions, among other factors.

Empirical studies that examine flood risk perceptions during a direct threat of a hurricane making landfall are limited. Exceptions are Meyer et al. (2014) and Botzen et al. (2022). Meyer et al. (2014) documented the dynamics of coastal residents' risk perceptions as Hurricane Isaac and Sandy approached the coast of Louisiana and New Jersey in 2012 using a real-time survey. Botzen et al. (2022) utilised a real-time hurricane survey approach at the end of the 2020 hurricane season to study the evacuation intentions and behaviour of coastal households in Florida. They compared these findings with evacuation intentions at the beginning of the hurricane season using a cross-sectional survey. Neither Meyer et al. (2014) nor Botzen et al. (2022) offered an analysis of the factors influencing flood risk perceptions, as is done in our study.

The objectives of our study are to understand the temporal dynamics in flood risk perceptions shortly before a hurricane makes landfall and afterwards, and to obtain insights into the factors that relate with these risk perceptions, including how they compare with objective indicators of the risk respondents faced at the time of the survey. Our study analyses data collected during the period in which Hurricane Dorian approached Florida in 2019 using a real-time survey. By resurveying part of the original sample a few months after the storm our paper also contributes to the flood risk perceptions literature by exploring these dynamics in the context of a near-miss hurricane event. Research on near-miss hurricanes has shown that people may underestimate the dangers of subsequent hazardous situations based on the experience of the near-miss, reasoning that the negative outcome did not materialize last time (Dillon et al., 2011; Dillon & Tinsley, 2016). These insights have been collected through vignette surveys, which are based on hypothetical scenarios. Our research goes beyond these previous studies by examining perceptions in response to a Category 5 hurricane predicted to make landfall in Florida. As such, the main innovation of our study is that we examine how various factors relate with dimensions of flood risk perceptions during an imminent threat of a hurricane as well as changes in these perceptions following an actual near-miss event.



The remainder of this paper is structured as follows: Section 2 provides a theoretical background and our
hypotheses about factors related to flood risk perceptions. Section 3 describes the survey and statistical methods.
Section 4 presents the results, and Section 5 discusses the key findings. Section 6 concludes.
**2.   Theoretical background**
Risk perceptions form an integral part of decision theories in behavioural economics and psychology, which
postulate that perceiving a high risk is a necessary condition for taking risk reduction actions (Kahneman &
Tversky, 1979; Hertwig & Wulff, 2022). Two thought processes that explains how people perceive and respond
to risks are System 1 and System 2 thinking (Kahneman, 2011). The former refers to an intuitive thinking process
that operates quickly, effortless and automatically. Furthermore, this mode of thinking has been associated with
heuristics. Heuristics refer to mental shortcuts that simplify the complex reality surrounding risks (Tversky &
Kahneman, 1973). By contrast, System 2 considers a more analytical risk assessment by evaluating the available
information more systematically and with more effort (Kahneman, 2011). For example, flood likelihood and
potential consequences are likely to be assessed by individuals based on information that is available to them.
Since individual perceptions of risk are expected to be shaped by System 1 and System 2, our explanatory
variables, as well as our hypotheses, are grounded in System 1 and System 2 thinking. We examine the influence
of experience, in line with the availability heuristic, and herding as part of System 1 thinking processes on flood
risk perception. The former refers to a type of cognitive bias in which an event's probability is evaluated based on
relevant examples that come to mind (Tversky & Kahneman, 1973). The latter, on the other hand, refers to the
mirroring of behaviour of other individuals. In the case of a highly uncertain or risky issues, individuals are more
likely to mirror behaviour (Kunreuther, 2021). The influence of actual risk and the development of Hurricane
Dorian on risk perception is analysed as part of System 2 thinking in our study, because accounting for such
information in one's judgement about risk takes considerable effort, in contrast to the heuristic-based judgements
that guide System 1 thinking processes.
**2.1  Heuristics (system 1)**
Consistent with the availability heuristic, a substantial amount of literature has found that previous experience with
a flood positively impacts the perceived flood probability as exposure to a flood may make the risk easier to recall
and more salient (Bradford et al., 2012; Peacock et al., 2005; Reynaud et al., 2013; Richert et al., 2017). Therefore,
we expect that past flood experience has a positive relationship with flood risk perceptions.
**H1**
Respondents who have experienced a flood have a higher perception of flood risk.
In addition to actual experience, and consistent with the availability heuristic, we argue that the perception of
specific characteristics and risks associated with a hazard, at one moment in time when the hazard is salient, may
make it cognitively easier to judge that similar experiences regarding the hazard and its associated risks in general
can occur in the future. In the case of Dorian, people faced the possibility of catastrophic damages and developed
risk perceptions, such as perceptions about the strength and severity of possible impacts. Individuals with high
perceptions of these specific hurricane characteristics may find future hurricane hazards, including their induced
flooding, easier to imagine. Thus, we expect high perceptions of specific hurricane characteristics (awareness of
living in a Dorian impact area and the perceived hurricane wind speed on the Saffir-Simpson Hurricane Wind
Scale) to increase perceived flood risk.
**H2**
Respondents with a high perception of specific Dorian characteristics have a higher perception of flood risk.
In a situation where individuals lack objective information regarding a hazard, they may dependent on local
government officials responsible for risk management instead. This might be the case in our context if people were
unaware of information on risk, or are unwilling to incur search costs associated with collecting information on
risk (Kunreuther & Pauly, 2004). Previous studies have found that individuals distrusting local government
officials in charge of flood risk management have a higher perception of risk regarding natural hazards (Siegrist
et al., 2005). Terpstra (2011) has shown that respondents who trust local risk management assess flood probabilities



as lower. Hence, we expect that trust in the capabilities of local government officials responsible for flood risk
management lowers flood risk perceptions.

**131 H3**

Respondents who have more trust the in the flood management capabilities of local government officials have a
lower perception of flood risk.
Few household survey studies have examined social factors as a driver of risk perceptions (Lechowska, 2018; Van
der Linden, 2015). We elicit the prescriptive dimension of social norms in our study (Cialdini et al., 1991).
Prescriptive social norms in the context of hurricane induced floods can be defined as the degree of social pressure
an individual feels to view floods as a risk that requires action (Van der Linden, 2015). It is hypothesised that
individual risk perceptions are amplified if social referents (friends, family, acquaintances) view an event as a risk
that should be acted upon (Swim et al., 2009).

**140 H4**

Respondents who acknowledge that important social referents (friends, family, acquaintances) believe that
someone in their (the respondent) situation ought to act upon the risk of floods have a higher perception of flood
risk.

### 144 2.2 Objective risk characteristics (system 2)

In line with System 2 thinking, previous studies have found a positive relationship between indicators of actual
flood risk and flood risk perception (Botzen et al., 2015; O'Neill et al., 2016; Richert et al., 2017; Rufat & Botzen,
2022). As such, we expect the flood probability at one's residence to be positively related to flood risk perception.
Furthermore, we expect that the floor of one's residency influences perceived flood risk, because those living on
lower floors are more exposed to flood water than people residing on upper floors (Lechowska, 2018). A similar
reasoning holds for people who reside in homes with a basement. Overall, we expect the presence of residence
characteristics that signal a high exposure to flooding, to be positively associated with perceptions of flood risk.

**152 H5a**

Respondents whose home is situated in an area with a high flood risk have a higher flood risk perception than
those whose home is situated in an area with a lower flood risk.

**155 H5b**

Respondents who occupy the ground floor at their home have a higher perception of flood risk than those who
live on an upper floor.

**158 H5c**

Respondents with a basement, cellar or crawlspace in their home have a higher flood risk perception than those
who do not have a basement, cellar or crawlspace in their home.
The flood risk caused by a hurricane making landfall varies as the characteristics of a hurricane develop over time
(Musinguzi & Akbar, 2021). Risk communication strategies regarding flood risk aim to raise awareness and
conform risk perceptions with the objective risk that residents face as the risk evolves (Kellens et al., 2013). In
the case of Hurricane Dorian, the National Oceanic and Atmospheric Administration (NOAA) informed
inhabitants in real-time, as the hurricane was approaching the coast of Florida, about the current level of hurricane
intensity. We expect high flood risk perceptions within periods in which the wind speed of the storm was high.
Furthermore, it has been observed that perceived risk, especially the sense of danger, is likely to decrease after a
near-miss of catastrophic damages (Baker et al., 2009). In the context of a near-miss situation, people may assume
that they escaped the danger and perceive the intervening good fortune as an indicator of resiliency (Dillon et al.,
2011; Tinsley et al., 2012). In addition, risk perceptions are likely to be high during the imminent threat of a
hurricane as flood risk is likely to be salient. As a result, we expect the level of worry and concern to decline
between the period during the threat of Hurricane Dorian and after the threat had dissipated.

**173 H6**



Respondents who finished the survey during time periods in which the maximum wind speed of Hurricane Dorian
was high have a higher flood risk perception.

**H7**

During a direct threat of a hurricane respondents have a higher flood risk perception compared to when this threat
has dissipated.

### 2.3 Personal characteristics

Besides heuristics and objective risk characteristics, personal characteristics such as risk preferences have been
identified as shaping risk perception (Feyisa et al., 2023; Villacis et al., 2021). In economic theories of decision
making, risk preferences/attitudes refer to the willingness of an individual to face a potentially risky situation
(Feyisa et al., 2023). Negative attitudes may result in an elevated view of risk levels, such as the probability of
loss (Prince & Kim, 2021). Therefore, we expect this personal characteristic to be positively associated with
perceived flood risk. Risk aversion is explicitly modelled as a determinant of risk perception, as implemented in
studies such as Cullen et al. (2018), Feyisa et al. (2023) and Villacis et al. (2021).

**H8**

Respondents who are risk averse have a higher flood risk perception than those who are risk seeking.
Locus of control may also be associated with risk perception (Breakwell, 2014; Ahmed et al., 2020). Locus of
control can be defined as an individual's belief about whether they have control over outcomes in their life (Rotter,
1966). People with an internal locus of control are of the opinion that their own efforts determine life outcomes in
their lives. In contrast, external locus of control types are of the opinion that these outcomes are out of their control
and often arise due to fate (Rotter, 1966). Since internal locus of control types may believe they have the propensity
to moderate their level of risk, e.g. by taking risk reduction measures, we predict that they are more likely to worry
less about risk than externals.

**H9**

Respondents who have a high internal local of control have a lower flood risk perception than those with an
external locus of control.

### 3. Methods

#### 3.1 Survey instrument and implementation

We conducted the real-time survey on the evening of August 29, 2019, till September 2, 2019. In total 871
responses were collected using telephone interviews. All participants are residents of Florida living in potential
flood areas based on the FEMA flood zone maps. The sampled respondents lived in neighbourhoods that were
forecasted to be potentially hit by Hurricane Dorian by the National Hurricane Centre (NOAA, n.d.). While the
projected path of Dorian remained uncertain during the five-day survey period, the survey sample was updated
over time to include areas where flood impacts were expected to be the largest. Figure 1 shows the geographical
distribution of survey respondents.



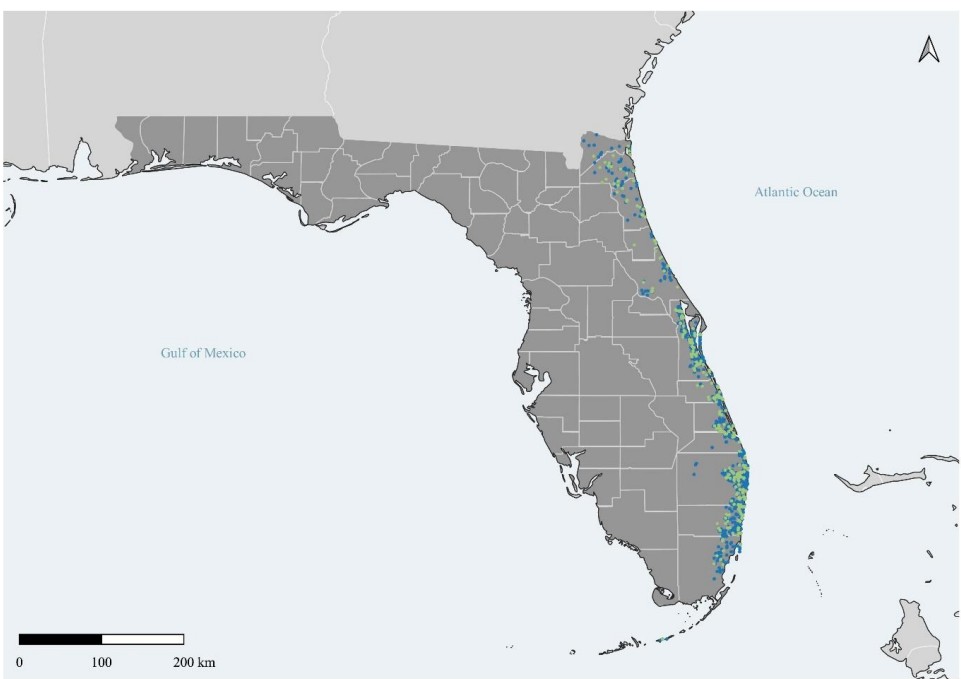

**Fig. 1** Locations of respondents in Florida in our initial survey (in blue dots) and follow-up survey (in green dots)

The second survey was administered several months after the near-miss of catastrophic damages from Dorian, among a subset of the first survey sample, in order to analyse how risk perceptions at the individual level changed after Hurricane Dorian. Particular care was taken to ensure similar sample characteristics across surveys in order to meaningfully compare samples in the analysis. Responses were collected using both phone interviews and online questionnaires. Participants who completed the second survey were offered a payment of 20 dollars. This amount was raised to 50 dollars to increase the survey response rate. Non-responders were reminded through a postal mail letter in which they were also informed of the monetary incentive. In total, 255 responses were collected. The sample's main socio-demographic characteristics are similar across the two surveys (see Table 1).

The gender distribution of the first survey is also comparable to that of the population of Florida. However, individuals over the age of 65 are overrepresented in the sample, as 49% of the respondents are 65 years and over compared to the 21% of citizens in Florida. Furthermore, the sample is skewed towards respondents with a college degree or higher (62%) compared to the Florida population (30%). Lastly, the median annual gross household income range is $75,000 to $124,999, which is higher than the $57,703 median household income after tax in 2018 in Florida (U.S. Census Bureau, n.d.).





**Table 1.** Socio-demographic characteristics of survey 1 and survey 2

| Variable | Sample survey 1 (871) | | Sample survey 2 (255) | |
|---|---|---|---|---|
| | Frequency | Percent | Frequency | Percent |
| *Gender* | *868* | | *254* | |
| Male | 416 | 47.93% | 128 | 50.39% |
| Female | 452 | 52.07% | 126 | 49.61% |
| | | | | |
| *Age (years)* | *809* | | *240* | |
| Mean (SD) | 62 (16.5) | | 62 (17.1) | |
| | | | | |
| *Education* | *849* | | *253* | |
| Some high school | 23 | 2.71% | 7 | 2.77% |
| High school graduate | 130 | 15.31% | 26 | 10.28% |
| Some college | 170 | 20.02% | 52 | 20.55% |
| College graduate | 325 | 38.28% | 96 | 37.94% |
| Post graduate | 201 | 23.67% | 72 | 28.46% |
| | | | | |
| *Household income 2018* | *663* | | *199* | |
| Less than $10,000 | 24 | 3.62% | 8 | 4.02% |
| $10,000 to $24,999 | 57 | 8.60% | 15 | 7.54% |
| $25,000 to $49,999 | 98 | 14.78% | 23 | 11.56% |
| $50,000 to $74,999 | 145 | 21.87% | 49 | 24.62% |
| $75,000 to $124,999 | 167 | 25.19% | 58 | 29.15% |
| More than $125,000 | 172 | 25.94% | 46 | 23.12% |


### 234 3.2 Measures

### 235 3.2.1 Dependent variables of general flood risk perceptions

A total of four measures were used to elicit subjective judgements about flood risk: two qualitative questions
regarding feelings about risk and two quantitative predictions of the flood probability and the cost to repair damage
in case of a flood. The coding of these variables can be found in Table S1 in the Supplementary Information. The
quantitative question regarding the flood probability asked respondents to judge the yearly likelihood that a flood
would occur at their homes on a logarithmic scale. Bruine de Bruin et al. (2011) and Woloshin et al. (2000)
observed that a logarithmic answer design performs well in eliciting perception of low likelihood risks.
Furthermore, we asked participants to indicate how worried they feel about the danger of a flood at their home, as
well as their feeling of concern about the consequences of flooding (following Botzen et al., 2015; Robinson &
Botzen, 2018; 2019).

### 245 3.2.2 Independent variables

A range of socio-demographic information was collected, including respondents' gender (1=female), age,
education, income and homeownership. These variables are included as control variables in our analysis.
One question was used to assess prior experience with flooding due to natural disasters. Respondents were asked
to recall how often their current home has been flooded during the time they had lived there. Responses were
dichotomised: 0 = no experience, 1 = at least one experience. To measure trust, we asked respondents to indicate
how much they feel they can trust the flood limiting capabilities of local government officials on a 4-point Likert
scale anchored from 1 = not at all to 4 = completely. Furthermore, we asked respondents two questions about the
extent to which they feel social pressure regarding the purchase of flood insurance and the implementation of risk
reduction measures on a 5-point Likert scale anchored from 1 = strongly disagree to 5 = strongly agree.
Two questions were used to assess Dorian specific risk perceptions. One question asked respondents to assess their
level of certainty that the area they live in will be affected by Hurricane Dorian. Respondents were also asked to
report the wind speed of Hurricane Dorian on the Saffir-Simpson Hurricane Wind Scale, based on the last time
they had received this information.



With regard to objective flood risk, three questions were asked to respondents to elicit the characteristics of their residence. Specifically, we inquired whether part of the building the participant occupies includes the ground floor level, and about the presence of a basement, cellar or crawlspace in the home. Furthermore, we gathered spatial information regarding objective flood risk using FEMA flood zone maps and respondents' zip codes. This information allowed us to geospatially classify the location of participants as either living within a 100-year flood zone (FEMA zone A) or outside of a 100-year flood zone.

Lastly, regarding individual preferences, both locus of control and risk preferences were elicited using a 10-point Likert scale. Respondents had to indicate how much they felt in control over their lives and how much risk, in general, they are willing to take. This qualitative survey question to elicit willingness to take risks in general has been shown to predict risk-taking behaviour across different contexts (Dohmen et al., 2011).

### 3.3 Statistical analysis

#### 3.3.1 Flood risk perceptions

Since the dependent variables are ordinal outcomes, we adopt ordered logistic regressions to assess the impact of independent variables on each of the flood risk perception dimensions. The ordinal nature of the dependent variables are accounted for using this method. Furthermore, regarding the interval distance of the answer options no assumptions are made (Liddell & Kruschke, 2018). For each independent variable the assumption of proportional odds applies, meaning that the coefficient estimate β is the same across logit equations for the different cut points (Fullerton, 2009).

A series of correlation tests of the explanatory variables were run to analyse multicollinearity. Taking 0.6 as a threshold value from the commonly recommend threshold range of 0.6-0.8 (Tay, 2017), social norms regarding risk mitigation and social norms regarding insurance were found to be highly correlated (r = 0.643). As a result, we created a new variable by synthesising the observations of these two variables (Cronbach alpha = 0.779) into one. The reason is that the high correlation implies that the two questions measure the same underlying construct, i.e. a tendency to comply with social norms.

#### 3.3.2 Change in flood risk perception

In order to analyse a potential change in the risk perception dimensions, during Hurricane Dorian and afterwards, change variables were calculated by subtracting the observations of the first survey from the observations of the second survey, for each risk perception dimension. Furthermore, logit regressions were performed for each change variable to examine determinants of change in perceptions of risk. The dependent variable $Y_i$ in the model is a dummy variable representing negative change (excluding positive change) or positive change (excluding negative change) in the risk perception of individual $i$, with the reference category indicating no change in risk perception. Independent variables were chosen for inclusion if they remained constant across individuals, in other words, if they were unaffected by the near-miss of Hurricane Dorian, namely: socio-demographic variables, residence characteristics, and flood experience. The socio-demographic and residence characteristics were only measured in the first survey, as significant changes were not anticipated.

#### 3.3.3 Flood risk misperception

Respondents were classified into groups that either underestimate, correctly estimate or overestimate risk. To do so, we compared the subjective valuation (SV) for the three different risk dimensions of each participant with the objective valuation (OV), allowing the error margins (EM) to differ according to previous studies regarding perceptions of flood risk (Botzen et al., 2015; Mol et al., 2020). Therefore, we consider the perceived risk estimate to be accurate when $OV(1 - EM) \leq SV \leq OV(1 + EM)$. The error margin for the perceived flood probability and hurricane wind speed is anchored at 0%, while the error margin for perceived flood damage caused by Hurricane Dorian is fixed at 50 %. The error margin of 0% was chosen for perceived flood probability and hurricane wind speed because the objective estimates, the FEMA flood zones and Saffir-Simpson Hurricane Wind Scale respectively, represent distinct categories. As a result, the estimates of respondents are either considered as correctly estimating the category, or not. The modelled flood damage data, on the other hand, is continuous and as such an interval was chosen for the error margin to reflect flood damage model uncertainty.

The objective flood damage was derived using a model cascade; first, the actual storm track of Hurricane Dorian was obtained from NOAA (Historical Hurricane Tracks, n.d.). The storm track was then translated into a spiderweb



format using 'Delft 3D' software that provides spatially explicit meteorological data, speed, and direction for the hurricane (Deltares, n.d.). The spiderweb data was used to force the Delft 3D Flexible Mesh to obtain inundation depths for all respondent locations. The inundation depths are all translated into a damage fraction by using HAZUS depth damage curves (FEMA, n.d.). Finally, by multiplying the reported value of the houses by the damage fraction, an objective estimate of flood damage is obtained per respondent.

In order to investigate the drivers of flood risk misperception, two logit regressions for each risk indicator were estimated. The dependent variable $Y_i$ in the model is a dummy variable depicting under-estimation (excluding over-estimation) or over-estimation (excluding under-estimation) of the risk dimensions of individual $i$. For all models the reference category is a correct estimation by the participants.

## 4. Results

### 4.1 Descriptive statistics of risk perceptions

During the first day of the survey the forecast indicated that Hurricane Dorian was predicted to make landfall in the middle of the east coast of Florida, with the uncertainty cone covering almost the entire state. Midway through the survey period landfall in Florida was still likely, but the hurricane was expected to turn away from the coast over time. On the last day of the survey, the predicted rightward shift became stronger (NOAA, n.d.). However, landfall in Florida was still within the cone of uncertainty. Furthermore, hurricane and flood warnings were issues along the coastline of Florida during the entire duration of data collection (NOAA, n.d.). As a result, respondents faced the threat of suffering flood damage from Hurricane Dorian during the entire time the survey was conducted.

It is notable that almost all participants had heard of the approaching hurricane (92%), of which the majority correctly indicated that Dorian was a hurricane (93%) instead of a tropical storm (6%). A small proportion of the sample stated that they did not know whether Dorian was a hurricane or tropical storm (1%). Nevertheless, 1 in 4 participants were unaware that they lived in an area that could be affected by the hurricane.

Moreover, almost all respondents in the second survey indicated that their primary source of information to stay updated about the approaching hurricane was the television (91%). In contrast, social media and face-to-face communication were less commonly utilised. Only 3% of respondents used Instagram or Twitter, while 18% of respondents used Facebook to gather information about Dorian. Respondents who followed specific social media accounts to acquire information about the storm, mainly followed the weather channel (14%).

In addition, there is a high perception of the flood probability among respondents (Table 2). 80% of respondents expect a yearly flood probability of 1/100 or higher at their home. Furthermore, the majority of the participants (81%) who live in the 1/100 flood zone reported a flood probability of 1/100 or higher, which shows that many respondents' flood risk perceptions align with the relatively high flood risk they face in reality.

**Table 2.** Comparison of actual and perceived flood probability

| Category of flood probability | FEMA flood zone A | | Total |
| --- | --- | --- | --- |
| | Yes | No | |
| N | 523 | 238 | 761 |
| More often than 1 in 10 years | 12.43% | 11.34% | 12.09% |
| Exactly 1 in 10 years | 19.69% | 22.27% | 20.50% |
| Between 1 in 10 years and 1 in 100 years | 15.68% | 17.65% | 16.29% |
| Exactly 1 in 100 years | 33.08% | 27.31% | 31.27% |
| Between 1 in 100 years and 1 in 1000 years | 3.25% | 1.26% | 2.63% |
| Exactly 1 in 1000 years | 4.40% | 8.40% | 5.65% |
| Less often than 1 in 1000 years | 11.47% | 11.76% | 11.56% |

However, this awareness does not result in feelings of concern about flooding, as a majority of respondents believed that the flood probability at their home is too low to be concerned about the consequences of a flood (54%). Similarly, the majority of the sample indicated that they strongly disagree or disagree with the statement "I am worried about the danger of a flood at my current residence" (59%) (Figure 2).



While the majority of the sample stated that they do not feel generally worried about the danger of a flood at their
residence, feelings of worry with regards to possible damage caused by Dorian specifically are present to a greater
extent. Only 28% of the respondents indicated that they strongly disagree or disagree with the statement concerning
feelings of worry about the hurricane causing damage to their home or home contents. As such, respondents were
more worried about damages caused by the approaching hurricane (65%) than flooding in general (36%).

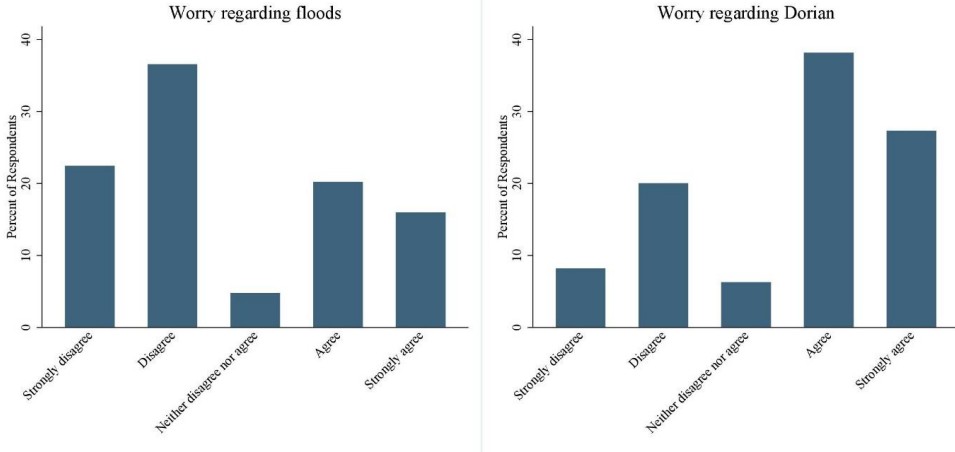


**Fig. 2** Distribution of responses to statements about worry of general flood damage (left) and damage caused by
Hurricane Dorian (right)
**4.2 Regression Analysis**
Flood risk perception is measured using four indicators in this study, namely worry about flooding, concern
regarding flood consequences, perceived flood probability, and the estimated cost to repair damage in case of a
flood. We present the results of the models for each dimension of flood risk in Table 3. Time fixed effects are
included in the estimations, but we suppress those coefficient estimates in the interest of conserving space.
Regarding socio-demographic variables, the predictor age is significantly correlated with worry about flooding.
The negative coefficient for age indicates that older people are less likely to be worried about the dangers of
flooding at their current residence compared to younger people. Moreover, the negative coefficient for completion
of some college indicates a lower damage estimate. Homeownership has statistically significant impact on
perceived flood probability and estimated damage.
We find a strong effect of flood experience and social norms across models. With the exception of estimated flood
damage, flood experience and social norms were found to be statistically significant in estimating the level of
worry, level of concern, and perceived flood probability. The positive coefficient on the flood experience variable
implies that those who have experienced flooding as a result of natural disasters are more likely to worry about
flooding, feel concerned about flood consequences at their home, and have a higher perception of the flood
probability compared to those who have not experienced flooding at their current residence. In addition, trust was
found to be negatively related with the level of concern. That is, those who trust the ability of government officials
to limit flood risk are less likely to feel concerned regarding the flood probability at their homes.
With the exception of worry, we find no effect for respondents' awareness of living in an area that was expected
to be affected by Hurricane Dorian on flood risk perception. Respondents who indicated that they were certain that
the area they live in is expected to be affected by Hurricane Dorian are more likely to feel worried about the
dangers of floods at their residence compared to respondents who were not sure whether they live in an area that
might be affected by the hurricane.



With regards to housing characteristics, the presence of a basement, cellar or crawlspace in one's house is
significantly related to the level of worry, but not to the level of concern, perceived flood probability and estimated
damage.
The regression models including the time fixed effects can be found in the Supplementary Information. Time
dummy variables, referring to the time and date within which respondents finished the survey categorized by when
maximum sustained wind speeds were published by the National Hurricane Centre, concerning the second and
third day of the survey period are significant in estimating levels of worry and concern. Participants who completed
the survey during time periods which have significant coefficient estimates have an increased likelihood of feeling
worried and concerned about the dangers and consequences of flooding compared to participants who completed
the questionnaire at the very beginning of the data collection.
Regarding the individual characteristics variables, we find no relationship between risk aversion and flood risk
perceptions, as well as between internal local of control and flood risk perceptions.





**Table 3.** Ordered logistic regression model of variables of influence on flood risk perception dimensions

| Variable | Worry | Concern | Estimated flood probability | Estimated flood damage |
|---|---|---|---|---|
| Age | -0.016* | -0.012 | -0.012 | -0.002 |
| | (0.007) | (0.006) | (0.008) | (0.007) |
| Gender | 0.174 | 0.179 | 0.155 | 0.283 |
| | (0.204) | (0.196) | (0.207) | (0.188) |
| Education | | | | |
|   - High school graduate | 0.905 | 1.734 | 0.873 | -1.220 |
| | (0.487) | (0.910) | (0.690) | (0.746) |
|   - Some college | 0.003 | 1.188 | 0.395 | -1.838* |
| | (0.470) | (0.887) | (0.682 | (0.758) |
|   - College graduate | 0.446 | 1.259 | 0.690 | -1.116 |
| | (0.480) | (0.890) | (0.681) | (0.717) |
|   - Post graduate | 0.391 | 1.251 | 0.695 | -1.201 |
| | (0.513) | (0.906) | (0.686) | (0.767) |
| Income | -0.071 | 0.075 | -0.063 | 0.163 |
| | (0.084) | (0.076) | (0.089) | (0.0923) |
| Home owner | 0.085 | -0.071 | -0.870* | 1.140** |
| | (0.352) | (0.376) | (0.409) | (0.393) |
| | | | | |
| Experience flooding | 0.854*** | 0.911*** | 1.683*** | 0.222 |
| | (0.273) | (0.271) | (0.299) | (0.240) |
| Social norms | 0.355*** | 0.331*** | 0.297*** | -0.071 |
| | (0.045) | (0.048) | (0.045) | (0.046) |
| Trust government | -0.135 | -0.213* | -0.109 | 0.033 |
| | (0.105) | (0.103) | (0.113) | (0.106) |
| | | | | |
| Awareness living in Dorian impact area | 0.291** | -0.020 | -0.077 | 0.153 |
| | (0.108) | (0.100) | (0.118) | (0.119) |
| Perceived wind speed Dorian | 0.034 | -0.041 | 0.019 | -0.012 |
| | (0.132) | (0.132) | (0.125) | (0.117) |
| | | | | |
| Home ground floor | -0.393 | -0.661 | -0.418 | 0.637 |
| | (0.396) | (0.391) | (0.458) | (0.388) |
| Basement | 0.721** | 0.288 | 0.006 | -0.264 |
| | (0.256) | (0.277) | (0.275) | (0.234) |
| FEMA flood zone | 0.076 | -0.126 | -0.051 | -0.095 |
| | (0.212) | (0.198) | (0.215) | (0.203) |
| | | | | |
| Risk aversion | -0.027 | -0.029 | 0.029 | 0.013 |
| | (0.034) | (0.034) | (0.039) | (0.035) |
| Internal locus of control | -0.052 | -0.015 | 0.003 | -0.022 |
| | (0.036) | (0.033) | (0.037) | (0.039) |
| | | | | |
| Log likelihood | -561.615 | -581.744 | -610.013 | -726.640 |
| Pseudo $R^2$ | 0.126 | 0.102 | 0.103 | 0.042 |
| Observations | 426 | 426 | 395 | 384 |

Notes: Time dummy variables are suppressed. Robust standard errors in parentheses. Significance levels:
*$p<0.05$; **$p<0.01$; ***$p<0.001$.

### 4.3 Differences in risk perception before and after the hurricane threat

Paired sample t-tests were performed to determine whether flood risk perceptions changed significantly during and after the threat of Hurricane Dorian. Most changes in flood risk perception are statistically insignificant, except for feelings of worry about the dangers of flooding. The mean decreased from 2.6 to 2.4 ($p=0.017$), suggesting that worry regarding flooding is higher during periods of extreme weather in line with our hypothesis.



With regard to the explanatory variables, all changes in personal beliefs and experiences are statistically
insignificant. Significant changes are observed for personal preferences variables. The mean of risk aversion
decreased from 3.9 to 2.8 (p<0.001). This implies that during the hurricane threat people were more risk averse,
which is not surprising in the context of an emergency situation. Feelings of control, on the other hand, slightly
increased. However, the change in means was not found to be statistically significant.

### 4.3.1 Exploratory regression analysis

Furthermore, we looked at potential predictors regarding the change in the risk perception dimensions (Table S3,
Supplementary Information). With the exception of flood experience and education, we find no effect of the
independent variables on the change of flood risk perception before and after Hurricane Dorian. Experience of a
flood increases the likelihood of feeling less worried and concerned about the dangers and consequences of a flood
at respondents' residence after Dorian. Respondents who have completed a higher level of education are less likely
to feel a lower level of concern about the flood consequences after Dorian.

### 4.4 Objective risk assessment

As can be seen in Table 4, the majority of participants overestimated the wind speed of the hurricane while it was
a Category 1 or 2 hurricane. Furthermore, the majority of respondents either underestimated or overestimated the
wind speed of Dorian while it was a Category 3 hurricane. As such, most of the misperceptions occurred while the
hurricane wind speed was low. In contrast, during the three day period in which Dorian developed into a Category
4 and 5 hurricane, the majority of respondents correctly estimated the wind speed of the storm. In total, 115
participants (16%) underestimated the wind speed of Hurricane Dorian, 511 participants (69%) correctly estimated
the hurricane category, and 110 participants (15%) overestimated the strength of Dorian.

**Table 4.** Distribution of hurricane wind speed estimates on the Saffir-Simpson Hurricane Wind Scale per day
(at 0% error margin)

|  | Category Hurricane Dorian | | | | |
|---|---|---|---|---|---|
|  | 1 | 2 | 3 | 4 | 5 |
| Underestimation | 0 (0.00%) | 12 (44.44%) | 30 (21.43%) | 47 (15.56%) | 26 (11.40%) |
| Correct within 0% error margin | 12 (30.77%) | 1 (3.70%) | 67 (47.86%) | 229 (75.83%) | 202 (88.60%) |
| Overestimation | 27 (69.23%) | 14 (51.85%) | 43 (30.71%) | 26 (30.71%) | 0 (0.00%) |

With regard to the perceived yearly flood probability at the residence of respondents, 423 (60%) participants
correctly stated that they live in an area with a flood probability of 1 in 100 years or less. In total, 287 participants
either underestimated or overestimated the probability of a flood. More precisely, 100 participants (14%)
considered the recurrence interval of a flood at their current residence as more than 1 in 100 years even though
they live in a 1 in 100 year flood zone, thereby underestimating the flood probability. A total of 187 (26%)
participants, on the other hand, overestimated the flood probability at their current residence, estimating the return
period as 1 in 100 years or less while living outside the FEMA flood zone A.

Figure 3 provides an overview of the distribution of under-, correct, and over-estimations for anticipated flood
damage. The vast majority of respondents, namely 356 participants (55%), overestimated the cost to repair the
damage of their home and its contents in the case of a flood.





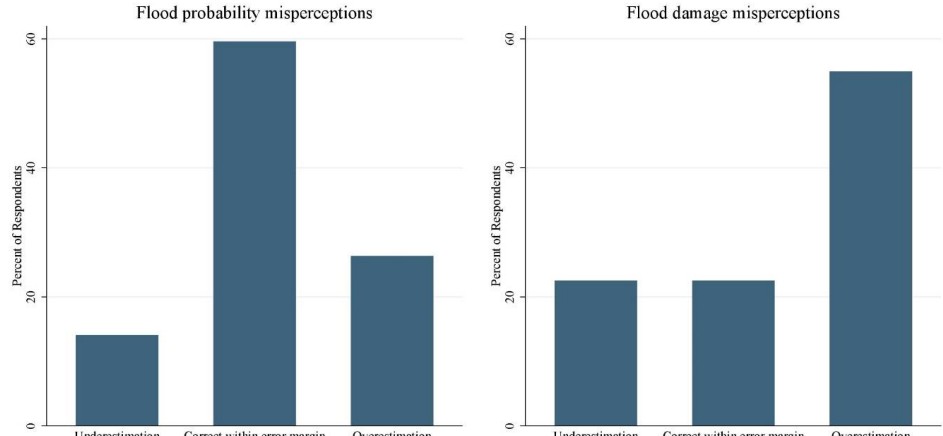

**Fig. 3** Distribution under-, correct, and over-estimations for anticipated flood probability (left, EM=0%) and
damage (right, EM=50%)

*4.4.1    Regression analysis*

Table S4 (Supplementary Information) reports regression results for the three dimensions of flood risk perception.
The negative coefficient for the variable concern indicates that respondents who perceive the flood probability as
sufficiently high to be concerned about the consequences of a flood are less likely to underestimate the flood
probability. In addition, those who are concerned are less likely to underestimate potential flood damage, while
those who are risk averse are more likely to overestimate the damage.

With regard to residence characteristics, the positive coefficient for occupation of the ground floor indicates that
individuals who live on the ground floor are more likely to overestimate the flood probability at their home. This
result makes sense, since individuals who live on the ground floor are more at risk regarding floods.

Regarding personal preferences, being risk averse makes it more likely that respondents will overestimate the cost
to repair their home and home contents in case of a flood. In other words, the more risk averse respondents are,
the more pessimistic they are in their estimation of the cost to repair the damage to their home caused by a flood.




**5. Discussion**
**Table 5.** Summary of hypotheses

| # | Description | Results | | | |
|---|---|---|---|---|---|
| | | Worry | Concern | Flood probability | Estimated damage |
| H1 | Respondents who have experienced a flood have a higher perception of flood risk. | S | S | S | NS |
| H2 | Respondents with a high perception of specific Dorian characteristics have a higher perception of flood risk. | PS | NS | NS | NS |
| H3 | Respondents who have more trust the in the flood management capabilities of local government officials have a lower perception of flood risk. | NS | S | NS | NS |
| H4 | Respondents who acknowledge that important social referents (friends, family, acquaintances) believe that someone in their (the respondent) situation ought to act upon the risk of floods have a higher perception of flood risk. | S | S | S | NS |
| H5a | Respondents whose home is situated in an area with a high flood risk have a higher flood risk perception than those whose home is situated in an area with a lower flood risk. | NS | NS | NS | NS |
| H5b | Respondents who occupy the ground floor at their home have a higher perception of flood risk than those who live on an upper floor. | NS | NS | NS | NS |
| H5c | Respondents with a basement, cellar or crawlspace in their home have a higher flood risk perception than those who do not have a basement, cellar or crawlspace in their home. | S | NS | NS | NS |
| H6 | Respondents who finished the survey during time periods in which the maximum wind speed of Hurricane Dorian was high have a higher flood risk perception. | PS | PS | NS | NS |
| H7 | During a direct threat of a hurricane respondents have a higher flood risk perception compared to when this threat has dissipated. | S | NS | NS | NS |
| H8 | Respondents who are risk averse have a higher risk perception than those who are risk seeking. | NS | NS | NS | NS |
| H9 | Respondents who have a high internal local of control have a lower flood risk perception than those with an external locus of control. | NS | NS | NS | NS |

Notes: S = supported , PS = partially supported, NS = not supported.



The results described in section 4 concerning our hypotheses are summarised in Table 5. Overall, flood experience and social norms are the most consistent predictor of flood risk perception. Various studies have observed the role experience plays in shaping flood risk perception (Bubeck et al., 2012b; Lechowska, 2018). In contrast, few papers discuss the role of socio-cultural context, which includes the influence of social norms, in relation to flood risk perceptions (Lechowska, 2018), which we find to be a key explanatory variable. Future studies on flood risk perceptions should include the socio-cultural context in order to approach flood risk perceptions in a more holistic manner.

The results are consistent with the availability heuristic (H1), in line with previous research (Bradford et al., 2012; Botzen et al., 2015; Peacock et al., 2005; Reynaud et al., 2013; Richert et al., 2017; Rufat & Botzen, 2022). Our assessment shows that the experience of a flood significantly and positively influences the flood risk perception dimensions of worry, concern, and perceived flood probability, but not estimated damage. The latter effect may be explained by the previously experienced floods not resulting in substantial damage. Furthermore, our findings provide additional insights to the literature on the availability heuristic in flood risk perception. We find that a direct flood experience influences flood risk perceptions to a greater extent than a high perception of specific hazard characteristics (H2). This result indicates that the experience of flooding matters regarding the availability heuristic, rather than being in a situation where the flood hazard is salient.

In addition, our findings do not strongly support the negative effect of trust on flood risk perceptions (H3). Previous research has suggested that higher levels of trust reduce perceptions of flood risk (Siegrist et al., 2005; Terpstra, 2011). While trust concerning government officials and their capability to limit flood risk negatively relates to concern regarding flood consequences in our study, we find no significant effect of trust on the other flood risk perception dimensions.

Social norms, on the other hand, are strongly related to risk perceptions. We find that social norms relate positively and significantly to worry regarding flooding, concern regarding flood consequences, and the perceived flood probability, confirming H4. Risk behaviour research in the context of flooding has found similar results (Lo, 2013; Poussin et al., 2014), indicating that individual uptake of flood risk reduction measures is amplified the more social referents recognize and act upon a risk. As such, our results add to the risk perception literature as social norms do not only influence the uptake of flood risk reduction measures, but are also associated with higher flood risk perceptions.

System 2 thinking processes, which include analytical risk judgements, are also found to influence risk perception. The positive relationship between objective and perceived flood risk is in line with previous literature (Botzen et al., 2015; O'Neill et al., 2016; Richert et al., 2017). With regard to residence characteristics, we find that the presence of a basement is positively related to the level of worry regarding flooding.

Furthermore, we find that the development of the hurricane forecasts concerning the hurricane wind speed has no impact on perceived flood probabilities. This finding suggests that the cognitive assessment of flood risk (flood probabilities) is largely insensitive to shifts in the maximum wind speed. In contrast, feelings about risk (worry and concern) are more susceptible to these changes. We find that worry and concern regarding floods are higher during periods in which the hurricane category is high.

Our data shows that after experiencing Hurricane Dorian, all dimensions of risk perception dropped. Previous studies have found similar results, demonstrating that people have a diminished risk perception after facing a near-miss natural hazard (Dillon et al., 2011; Dillon & Tinsley, 2016). However, the current analysis finds only partial support for H7, as worry was the only variable to decrease significantly after Hurricane Dorian. Regarding the explanatory variables, we find a significant decrease in risk aversion after the near-miss of Hurricane Dorian. The decline of risk aversion suggests that in the context of natural hazards risk preferences vary over time, with individuals being more risk averse during a direct threat and less risk averse following a near-miss, rather than being a stable personality trait (Schildberg-Hörisch, 2018).

With regard to the over- and under-estimations of risk dimensions, many respondents have accurate perceptions of the risks they face. Most respondents correctly recalled the maximum wind speed of Hurricane Dorian, especially when it was high (Category 4 of 5), but mis-estimated it when the wind speed was low (Category 1 or 2). These results may indicate an enhanced communication of, or interest in, the risk as Dorian proceeded to rapidly intensify by September 1. Similarly, most of the respondents correctly perceived the flood probability at their homes. The overall correct estimation of the flood probability is in contrast to some previous work (Botzen et al.,



2015; Mol, 2020). Floods are much more frequent in Florida compared with the areas focused on in these previous
studies, which may explain a more rational appraisal of the flood probability in Florida. Regarding the estimated
damage, more respondents overestimated (55%) than underestimated (23%) the cost to repair damage in case of a
flood. The results show that being risk averse contributes to this overestimation. Respondents who think that the
flood probability is above their threshold level of concern, on the other hand, are less likely to underestimate the
cost of repairing the damage to their home and home contents in case of a flood. This result is consistent with the
findings of Botzen et al. (2015), who found that individuals who assessed the flood probability to be below their
threshold level of concern are more likely to underestimate their flood damage.

### 5.1  Policy implications

We found that during a direct threat of a hurricane, in which risk of flooding is high, individual risk perceptions
are high as well. However, misperceptions still prevail. 1 in 4 participants incorrectly perceived themselves as
living in an area that could not be impacted by Hurricane Dorian. Furthermore, we find that most people over-
estimated the wind speed of Hurricane Dorian when it was low (Category 1 or 2). With regard to damage
perceptions, most people overestimate the cost of repairing damage in case of a flood. Taken together, these results
regarding misperceptions show the importance of improving risk communication strategies, especially in cases
where risk perceptions are significantly lower than objective risk. Risk communication during the storm can be
improved by spreading more information about the storm and the areas it can affect to the inhabitants of these
areas. Furthermore, we find that flood risk perceptions are high during an imminent hurricane threat. Periods in
which risk perceptions are more likely to be high are suitable moments to motivate and inform people about
appropriate dry and wet flood-proofing measures using risk communication campaigns (Botzen et al., 2020;
Bubeck et al., 2012b). Therefore, communication policies during a hurricane threat should not only focus on the
risk itself, but also on the risk reduction measures people can implement during times of heightened risk
perceptions.
Based on our result, we recommend that raising awareness and activating social norms should be the focus of these
campaigns. The decline in worry regarding the dangers of a flood in combination with the strong influence of
previous flood event experience on flood risk perception highlights the need to preserve the memory of past floods.
Enlisting the help of those whom inhabitants feel trust for or trust as experts could lead to employing the most
influential sources in the communication of flood risk information. However, the effectiveness of activating social
norms depends on the careful design of communication messages and is highly context dependent (Bicchieri &
Dimant, 2022; Hauser et al., 2018).
Moreover, promoting flood risk awareness in the absence of a natural disaster is especially important after a near-
miss hazard, since our findings show that risk perceptions decline after the near-miss. The uniqueness of each
storm should be stressed in communication strategies, with the possibility of a direct hit for each hurricane being
taken serious in order to prevent the underestimation of flooding caused by natural disasters.

### 6.  Conclusion

Flood damage caused by hurricanes is predicted to continue to increase in the future. Flood preparedness and
support of flood risk management policies among the public are needed to reverse this trend. However, empirical
studies on household preparedness show that many households are underprepared for hurricane induced floods,
which to a larger extent could be due to low flood risk perception. We investigated various determinants of flood
risk perceptions and aimed to understand flood risk misperceptions of coastal residents in Florida in order to give
recommendations for flood risk communication strategies.
The novelty of our approach can be considered the main addition to the literature, as we employed a real-time and
follow-up survey during and after the threat of Hurricane Dorian. The former allows for a relatively unique and
important understanding of flood risk perceptions and their drivers during a period in which the hurricane threat
is heightened, while the latter provides a longitudinal view of the change in risk perceptions after the close call of
Hurricane Dorian making landfall in Florida.
Overall, the results show that while there is a high awareness of the flood probability, this awareness does not
necessarily translate into a high concern or worry about flooding. However, participants tended to perceive the
approaching hurricane as more of a threat with regard to the possible damage caused by Dorian. Still, 1 in 4
participants were unaware that they were living in an area that was predicted to be impacted by Hurricane Dorian.



After the near-miss, participants indicated that they felt less worried regarding the dangers of flooding and risk aversion declined.

Regarding the drivers of the flood risk perceptions, we find that previous flooding experience, in line with the availability heuristic, and social norms have the most consistent influence[4]. Furthermore, we observe a significant relationship with variables representing System 2 thinking, although to a lesser extent than the System 1 processes.

Based on our results, the following policy recommendations can be drawn. Information campaigns should aim to preserve the memory of past floods among the population, as well as focus on activating social norms. Furthermore, the observation that worry regarding the dangers of flooding declined after a near-miss shows the importance of regular campaigns promoting risk awareness after a near-miss. In order to prevent the underestimation of flooding caused by hurricanes, each possibility of a direct hit should be taken seriously.



## Data availability

The raw and processed data are not publicly available as the participants of this study did not give written consent for their data to be shared publicly.

## Competing interest

The authors declare that they have no conflict of interest.

## Financial support

This research was funded by the State of Florida Division of Emergency Management.



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
