# Peer review of "Factors of influence on flood risk perceptions related to Hurricane Dorian: an assessment of heuristics, time dynamics and accuracy of risk perceptions"

_Natural Hazards and Earth System Sciences, 2023_

## Author Response (AR1)

**Reply to reviewers**

We sincerely thank both reviewers for their useful comments and suggestions. We hope that the editor and the reviewers find our answers satisfactory.

The original comments of the referees are given in italics, followed by our response in normal font. Potential adjustments to the text in the manuscript are given in bold.

*Reviewer comments:*

*Reviewer 1 (remarks to the author)*

*The paper is interesting and approaches a very important topic. Notwithstanding it needs some major adjustments to be published.*

> We thank the reviewer for acknowledging the interest of our paper and the importance of the topic it addresses, as well as for their detailed suggestions for improvements. In each reply we explain how we will address the comment in our revised manuscript.

*Comments:*

1. *Theoretical background: maybe, when describing the heuristics, it should be helpful for clarity to report the hypotheses before the background for their formulation.*

   > We would like to thank the reviewer for their suggestion. We chose to first describe the literature background before formulating the hypotheses in order to enhance the flow of the text. The reason is that in this order, our hypotheses logically follow from the description of the heuristics. We will clarify this structure upfront in Section 2.

2. *Method section: In this section an explanation of how have been collected the interviews is needed, because it is only reported the number. Maybe before they are all the residents of the study area? In this case is also necessary to report it. How have been chosen the individuals for the 255 interviews after the event?*

   > We will address this comment by including additional information on the selection of participants. With regard to the second comment, all participants from the first survey round were invited to participate in the second survey round after the hurricane event. In total, 255 participated in the second survey. This paragraph will be adjusted to avoid confusion.

   > Adjusted paragraph: "The real-time survey was conducted from the evening of August 29, 2019, till September 2, 2019. In total 871 responses were collected using telephone interviews. **The interviews were administered by the company Downs and St. Germain, had a response rate of 12% and lasted 20 minutes on average**. All participants are residents of Florida living in potential flood areas based on the FEMA flood zone maps."

   > Adjusted paragraph: "The second survey was administered several months after the near-miss of catastrophic damages from Dorian **among the first survey sample**, in order to analyse how risk perceptions at the individual level changed after Hurricane

Dorian. Responses were collected using both phone interviews and online questionnaires."

3. *Measures: 3.2.2 Independent variables – as in the 3.2.1 paragraph it should be better to describe the variables and to recall the supplementary material for the coding.*

We will remove the references to coding in this section in line with 3.2.1 and instead refer to the supplementary information.

4. *Statistical analysis: this is the most critical section (sub-sections included). It is not easy to read and mostly unclear. I understand that the authors wanted to avoid an excessive use of technical terms, but the result is that section is not understandable per-se, without reading also the cited references. The cited references are needed in case the reader would like to deepen the subject, but the main text must convey clear information. I suggest to completely rewrite this section, to make it self-sustainable at least at a basic level.*

We acknowledge the concerns about the clarity of this section and the importance of making the text accessible without delving into the cited references. To address this issue we will undertake a comprehensive revision of this section to ensure that it is self-sustainable at the basic level, as well elaborate on the statistical analysis itself to deepen the subject.

5. *Results: It is unclear why for similar data have been used different representations: reported in the text, or in a table, or in bar chart. Maybe a deeper consideration about what type of data representation it is necessary. Furthermore, the Tables need to be graphically adjusted; for example, the first column of Table 2 needs to be enlarged in order to have the same row number in all the columns. The same problem interests other Tables, then I will not report again the same suggestion for other section. Maybe some table could be better shown in a horizontal layout, for example? When reported the difference in risk perception before and after the event it is mentioned the use of a t-test that it is not mentioned in methods.*

Tables 1, 2, 3 and 4 provide additional information to the reader which is not discussed in the text as it is not critical to supporting the main argumentation, but does provide the full scale of information. The information contained in these tables are more intuitive to interpret in a table than in a figure. Therefore, we have chosen to retain these tables. All tables will be graphically adjusted according to the suggestion of the reviewer.

With regard to Figure 2 and 3, we choose to represent the distribution of data in a bar graph to provide the reader with a visual asset that captures the data in addition to the text. We will adjust the figures to make the comparison of the data more comprehensive.

We will include the t-test in the methods.

6. *Are discussed data reported in the Supplementary materials, that should be better placed in the Result main paragraph.*

We included Table S3 and S4 in the supplementary information with the aim to conserve space. We will clarify the placement of the tables in the Supplementary Information upfront in section 4.3.1 and 4.4.1

7. *Discussion: at the beginning of the Discussion it is reported a table that it is a result, indeed. It could be better to move such table at the end of the Results section. This shift does not prevent the Discussion to be started almost in the same manner. In the discussion are mentioned suggestion for future studies that should be better included in the conclusion.*

> We chose to include Table 5 in the this section as the aim of the discussion section is to provide a discussion of our results related to the hypotheses, whereas the result section does not explicitly refer to the hypotheses. As we test quite many (nine) hypotheses, we thought it would be useful to provide the reader with a summary to give them an accessible overview whether the results prove or disprove the hypotheses. We will also include the subtitle hypotheses to reinforce the aim of this section.

> We will include suggestions for future studies in the conclusion instead of the discussion.

8. *Policy implications: the section Policy implications seems a mix between a further piece of the Discussion section and a Conclusion piece. Maybe it could be useful to eliminate this sections and to use its contents to enrich both Discussion and Conclusion. Also the Conclusion of an article must be almost self-consistent, then the sentence at row 554 needs to be extended, explaining only a bit more what is System 2 and what System 1.*

> We reason that the policy implication section might seem like a mix between a discussion and a conclusion section as we first summarize the results to show the importance of improving strategies regarding risk reduction. We will rewrite the text so that the focus of this paragraph is on the policy recommendations, while line 509 to 513 will be used to enrich the conclusion.

> Adjusted paragraph: "**Our results show that misperceptions prevail.** 1 in 4 participants incorrectly perceived themselves as living in an area that could not be impacted by Hurricane Dorian. Furthermore, we find that most people over-estimated the wind speed of Hurricane Dorian when it was low (Category 1 and 2). **These misperceptions show the importance of improving risk communication strategies.** Risk communication during the storm can be improved by spreading more information about the storm and the areas it can affect to the inhabitants of these areas. Furthermore, we find that flood risk perceptions are high during an imminent hurricane threat. Periods in which risk perceptions are more likely to be high are suitable moments to motivate and inform people about appropriate dry and wet flood-proofing measures using risk communication campaigns (Botzen et al., 2020; Bubeck et al., 2012b). Therefore, communication policies during a hurricane threat should not only focus on the risk itself, but also on the risk reduction measures people can implement during times of heightened risk perceptions.

> We will elaborate on System 1 and System 2 in line 554 to improve the conclusion in line with the suggestion: "Furthermore, we observe significant relationships between

variables **associating with the mode of thinking that represents the deliberate and analytical mental process (System 2) and perceived flood risk**, although to a lesser extent **than variables associated with the intuitive thinking process that operates quickly and automatically (System 1).**"

9. *Supplementary materials: in the caption of Table S1 should be specified the type of ranking (e.g. Likert scale)*

   We will adjust Table S1 accordingly.

10. *Final considerations: all the variables, all over the article should be written to be immediately recognizable in the main text, then it could be useful to use Capital letters for the initials, Italic for the name etc. For the other tables applies what written in C5.*

   We will address this comment by using italics to indicate the variable names throughout the manuscript.

*Reviewer 2 (remarks to the author)*

*The authors have presented a well-written manuscript describing survey results for flood perceptions collected during and after a hurricane warning event. The authors conclude with policy recommendations to improve risk communications based on their survey outcomes. I only have a few minor comments for further improving the manuscript.*

We would like to thank the referee for their kind words and attention to details.

*Comments:*

1. *Line 123 - ...they \*may dependent\* on...*

    This will be corrected to \***may depend**\* on.

2. *Paragraph 189 - Instead of referring to people as "internal/external locus of control types", suggest referring to them as "people with higher internal/external locus of control". This would prevent the perception of defining individuals as their type, and instead describing them as having a certain affinity.*

    We will adopt the reviewer's suggestion for this sentence.

3. *Line 197 - ...\*local\* of control*

    This will be corrected to \***locus**\* of control.

4. *Paragraph 218 - Fig 1 shows that the surveys were collected from coastal communities in Florida but the demographic comparison is done against the entire state. It would be more representative to perform the demographic comparison at zipcode or county levels corresponding to the survey respondents.*

    We will take the comment of the reviewer into account and perform a demographic comparison at the county level. The paragraph will be rewritten as follows:

    "The gender distribution of the first survey is comparable to that of the population of the **coastal counties**. However, individuals over the age of 65 are overrepresented in the sample, as 49% of the respondents are 65 years and over, **compared to the 25% for actual citizens of the coastal counties in Florida**. Furthermore, the sample is skewed towards respondents with a college degree or higher (62%), compared **to the coastal population (33%)**. Lastly, the median annual gross household income is **$100,000 in our sample, compared to the $70,331** median household income **of the coastal counties** after tax in 2018 (U.S. Census Bureau, n.d.)."

5. *Line 278 - ...the commonly \*recommend\* threshold range...*

    This will be corrected to \***recommended**\* threshold range.

6. *Line 423 - The statement "...considered the recurrence interval of a flood at their current residence as more than 1 in 100 years..." seems to indicate that respondents estimated the recurrence to be more frequent than 1 in 100 years (let's say, 2 in 100, etc.). In that case, wouldn't that align with or overestimate the flood probability if they lived in a 100-year flood zone?*

The reviewer is correct that the current word choice seems to indicate that the respondents are overestimating their flood probability.

In line with the reviewer's comment this paragraph will be rewritten as follows: "With regard to the perceived yearly flood probability at the residence of respondents, 423 (60%) participants correct stated that they live in an area with a flood probability of 1 in 100 years or higher. In total, 287 participants either underestimated or overestimated the probability of a flood. More precisely, 100 participants (14%) considered the recurrence interval of a flood at their current residence **as less frequent than 1 in 100 years even though they live in FEMA flood zone A**, thereby underestimating the flood probability. A total of 187 (26%) participants, on the other hand, overestimated the flood probability at their current residence, estimating the return period as 1 in 100 years **or more frequent** while living outside **the 1 in 100 years flood zone**."

7. *Line 426 - Statement will be easier to interpret if it was modified to "...estimating the return period as 100 years or less...", since that would directly imply a less than 100-year return period, thus overestimating the probability.*

    In our reply to comment 6 we have proposed an adjustment to this paragraph which also addresses this comment.

8. *Few references are missing publication year, and are cited as n.d. I would defer to the journal guidelines and the editor whether that is acceptable.*

    We will adjust the references which are referred to in this comment according to the journal guidelines.